# Oral Ingestion of Bean Sprouts Containing the HASPIN Inhibitor Coumestrol Increased Blood Testosterone Levels in Men

**DOI:** 10.3390/biology14080907

**Published:** 2025-07-22

**Authors:** Keisuke Kakazu, Akira Tsujimura, Yasushi Miyagawa, Kentaro Takezawa, Kazuhiro Kobayashi, Ryoji Yoshimura, Kensuke Nakajima, Seitaro Kamiya, Hiromitsu Tanaka

**Affiliations:** 1Laboratory of Molecular Biology, Faculty of Pharmaceutical Sciences, Nagasaki International University, 2825-7 Huis Ten Bosch, Sasebo 859-3298, Nagasaki, Japan; k.kakazu@niu.ac.jp; 2Department of Urology, Juntendo University Urayasu Hospital, 2-1-1 Tomioka, Urayasu 279-0021, Chiba, Japan; atsujimu@juntendo.ac.jp; 3Department of Urology, The University of Osaka Graduate School of Medicine, 2-2 Yamadaoka, Suita 565-0871, Osaka, Japan; miyagawa-yasushi@sumitomo-hp.or.jp (Y.M.); takezawa@uro.med.osaka-u.ac.jp (K.T.); 4Department of Urology, Sumitomo Hospital, 5-3-20 Kita, Osaka 530-0005, Osaka, Japan; 5D Clinic Tokyo, 1-11-1, Chiyoda 100-6210, Tokyo, Japan; k.kobayashi@d-clinicgroup.jp; 6Department of Health and Nutrition, Faculty of Health Management, Nagasaki International University, 2825-7 Huis Ten Bosch, Sasebo 859-3298, Nagasaki, Japan; yoshimurar@niu.ac.jp; 7Department of Pharmacy, Faculty of Pharmaceutical Sciences, Nagasaki International University, 2825-7 Huis Ten Bosch, Sasebo 859-3298, Nagasaki, Japan; naka@niu.ac.jp; 8Department of Pharmaceutics, Faculty of Pharmaceutical Sciences, Nagasaki International University, 2825-7 Huis Ten Bosch, Sasebo 859-3298, Nagasaki, Japan; kamiya@niu.ac.jp

**Keywords:** late-onset hypogonadism, longevity, health promotion, nutrition, polyphenol, isoflavone

## Abstract

Late-onset hypogonadism (LOH), characterized by age-related declines in testosterone levels, is associated with the development of various diseases. We previously demonstrated that the oral intake of bean sprouts rich in the HASPIN inhibitor coumestrol increased blood testosterone levels in mice. In this study, we showed that consuming bean sprouts also elevated blood testosterone levels in humans. These findings suggest that coumestrol-rich bean sprouts may serve as an effective dietary intervention for managing LOH.

## 1. Introduction

The advances in medical technology, sanitation, and living conditions have led to increases in human lifespan. This increased longevity has been accompanied by changes in patterns of disease, with cancer and dementia now among the most common causes of death. Late-onset hypogonadism (LOH), a condition caused by a decline in testosterone level with age, contributes to the onset of various diseases and represents an important target for extending the healthy lifespan of different populations [1]. LOH syndrome is a form of hypogonadism not attributed to chromosomal or genetic abnormalities, testicular injury, tumors, functional impairments, or central nervous system disorders. It primarily results from an age- or stress-induced decline in testosterone levels, with an age-related decrease in testosterone levels being the predominant cause. Treatment for LOH typically includes Chinese herbal medicine and testosterone replacement therapy (TRT), along with lifestyle modifications [2]. Testosterone is primarily produced by Leydig cells in the testes, with minor contributions from the adrenal cortex. The age-associated decline in testosterone levels is believed to result from reduced production and secretion by Leydig cells. In our previous work, we demonstrated that the intraperitoneal administration of CHR-6494, an inhibitor of the serine–threonine kinase HASPIN, in *Apc^Min/+^* mice—a model of familial colorectal cancer—inhibited both the development of colorectal cancer and testicular atrophy due to impaired spermatogenesis, while restoring blood testosterone levels [3]. Moreover, we found that the oral administration of coumestrol, a natural HASPIN inhibitor, suppressed Alzheimer’s disease onset in 5XFAD mice, a model of the disease, and elevated serum testosterone levels in both these mice and wild-type controls [4]. These findings suggest that HASPIN plays a role in testosterone production and secretion by Leydig cells.

HASPIN is highly expressed in haploid germ cells and is also present at low levels in various tissues, including in the brain [5]. It has been suggested that it plays important roles in mitotic cells by phosphorylating histone H3 at threonine 3 (Thr3), thereby facilitating the centromeric localization of the chromosomal passenger complex, a process that is essential for accurate chromosome segregation during cell division [6]. Functional analyses in plants and nematodes have further shown that HASPIN plays multiple roles beyond chromosome segregation [7,8]. The KINOME project identified numerous protein kinase inhibitors, including HASPIN inhibitors, and the inhibition of HASPIN has been shown to suppress the proliferation of cancer cells [9]. Several HASPIN inhibitors have since been characterized, and their antiproliferative effects against a wide range of human cancer cell types have been reported, including colon, breast, prostate, skin, pancreas, lung, ovary, bladder, biliary tract, and hematopoietic system cancers [10]. Similar anticancer effects have also been observed following the RNA interference (RNAi)-mediated knockdown of HASPIN expression [11]. No major side effects have been reported in HASPIN-deficient mice, or in preclinical studies of HASPIN inhibitors, suggesting that small-molecule compounds targeting HASPIN may be promising candidates for cancer therapy [12]. Coumestrol, a natural compound found in bean sprouts, has been reported to inhibit HASPIN kinase activity and suppress cancer cell proliferation [13].

Alzheimer’s disease, in which testosterone has been implicated, is characterized by the accumulation of amyloid beta in the brain and the subsequent phosphorylation of tau in neurons. Previously, we showed that HASPIN phosphorylates tau [4]. We have also established a method for cultivating bean sprouts rich in the HASPIN inhibitor coumestrol [14], and showed that the oral administration of these bean sprouts to 5XFAD mice, a widely used animal model of Alzheimer’s disease, reduced the levels of phosphorylated tau and amyloid beta in the hippocampus and prevented short-term memory loss [4]. It was also shown that the ingestion of bean sprouts increased blood testosterone levels, reduced visceral fat, and increased the length of the intestine in mice [15]. This intestinal elongation was suggested to increase the absorption of arginine and citrulline produced by intestinal bacteria, which, in combination with increased blood testosterone levels, may have an anti-aging effect [16].

To clarify the effective dosage of orally ingested coumestrol-rich bean sprouts in humans, we monitored changes in blood testosterone levels, a readily measurable in vivo indicator. Our results showed that the consumption of bean sprouts led to an increase in blood testosterone level. These findings suggest that the dietary intake of bean sprouts may alleviate the symptoms of LOH and may exert protective effects against other aging-related diseases.

## 2. Materials and Methods

### 2.1. Study Setting and Participants

Nine male volunteers aged between 57 and 77 years from Sasebo City were recruited to this study. All participants were healthy, with no underlying medical conditions and no history of use of medications or dietary supplements. The participants provided informed consent for the consumption of bean sprouts and for venous blood collection prior to bean sprout ingestion, and again after 1 and 3 months for the measurement of serum testosterone levels. Blood sampling was performed under medical supervision. There were no adverse events in cases requiring medical intervention. This study was conducted in accordance with the Declaration of Helsinki. All procedures involving human subjects were approved by the Institutional Research Ethics Committee of Nagasaki International University (approval no. 75).

### 2.2. Bean Sprout Capsules

Bean sprouts (*Vigna radiata*) containing 14 mg of coumestrol per 100 g of fresh weight were cultivated as described previously [14]. The sprouts were dried overnight at 75 °C and powdered using a mixer. Cellulose capsules were filled with 250–300 mg of the resulting bean sprout powder (Qualicaps Co., Ltd., Nara, Japan). Analysis by a certified food analysis center confirmed that the capsules contained no detectable bacteria or aflatoxins (Figure 1).

### 2.3. Regimen

The participants took 10 capsules orally every day for 3 months at a time convenient to them, corresponding to a daily intake of 2.5–3.0 g of bean sprout powder. Samples of approximately 2 mL of venous blood were collected from each participant between 9:00 and 10:30 a.m. at two time points: prior to the start of capsule ingestion, and 3 months after starting ingestion. No capsules were taken on the day of blood collection.

### 2.4. Testosterone Measurement

Blood samples were stored at 4 °C for 1 day, after which the serum was separated by centrifugation at 15,000× *g*. Testosterone was measured in aliquots of 20 μL of serum by ELISA using a commercial kit in accordance with the manufacturer’s protocol (Testosterone High Sensitivity ELISA kit [ADI-900-176]; Enzo Life Sciences, Farmingdale, NY, USA). The detection sensitivity of the assay was 2.6 pg/mL, with an average coefficient of variation of 3.3% across sample concentrations. Each sample was analyzed in duplicate using two assay blocks simultaneously. If a significant discrepancy between the two measurements was observed, the assay was repeated for that sample.

### 2.5. Statistical Analysis

Data are expressed as mean ± standard deviation. The normality of paired differences was assessed using the Shapiro–Wilk test. Differences between the baseline and the 3-month measurements were analyzed using the Wilcoxon signed-rank test. *p*-values < 0.05 were considered statistically significant.

## 3. Results and Discussion

The results are presented in Figure 2. The mean serum testosterone level among the participants prior to bean sprout ingestion was 3708 ± 1151 pg/mL. The serum testosterone level had increased in eight out of nine participants compared with the baseline, with an average value of 5209 ± 1876 pg/mL after ingestion. Although a placebo-controlled trial was not conducted, a previous study showed no placebo effect on blood testosterone level associated with the oral ingestion of other foods [17]. Given that eight out of nine participants showed an increase in testosterone levels, it is unlikely that the observed effect was due to a placebo alone. The participants were asked to consume the bean sprouts without making any specific lifestyle changes. One of the nine subjects experienced a decrease in testosterone levels after three months, which may have been influenced by factors such as physical activity, diet, sleep, or alcohol intake prior to blood sampling. Therefore, the observed increase in blood testosterone level was attributed to the oral ingestion of bean sprouts rather than any placebo effect. Although this small-scale study in elderly individuals demonstrated a significant increase in testosterone levels, larger studies involving a broader age range are needed to further clarify the testosterone-boosting effects of bean sprouts and to identify the target populations most likely to benefit.

Testosterone exists in three forms in the blood, albumin-bound testosterone (approximately 25–65%), sex hormone-binding globulin-bound testosterone (SHBG) (approximately 35–75%), and free testosterone (approximately 1–3%), with free testosterone being the most biologically active [18]. Total testosterone levels can vary depending on the assay used, due in part to the differences in the sensitivity to protein-bound forms [19,20]. Although testosterone subtypes were not assessed separately in this study, the total testosterone levels measured in the present study using a commercial ELISA kit were consistent with those reported previously by Iwamoto et al., who measured total testosterone in Japanese men by radioimmunoassay (RIA) [19].

To account for diurnal variation, blood sampling was performed between 9:00 and 10:30 a.m. [21]. The testosterone levels in collected blood samples were stable, and the storage method used in this study did not influence the interpretation of the measurements [19]. All observations were made between January and March, during which seasonal fluctuations in male testosterone levels are minimal; they may even decrease slightly [22,23]. Therefore, the observed changes in testosterone concentrations far exceeded expected variations due to time of day or season, supporting the conclusion that the oral ingestion of bean sprouts was responsible for the increase in testosterone levels. The assay was performed using a standardized method with a commercial kit, further supporting the reliability of our findings.

Soybeans and bean sprouts contain various polyphenols; however, unsprouted soybeans contain very little coumestrol, which is found mainly in bean sprouts [14]. Coumestrol is a polyphenol phytoestrogen that has been shown to bind to estrogen receptors [24]. Phytoestrogens, including coumestrol, can exert biological effects even at very low concentrations and are categorized as natural endocrine-disrupting chemicals [25]. Although it possesses a steroid ring, it has not been reported to interact directly with androgen receptors. However, it may act selectively on human sex steroid hormone receptors or influence the steroid hormone metabolic system, thereby contributing to increases in testosterone directly or participating in hormone regulation. Furthermore, coumestrol may regulate testosterone production via multiple pathways. Although the phytoestrogenic effects of polyphenols from soy-based foods have been documented, no prior studies have demonstrated an increase in testosterone levels associated with their intake. Therefore, it is also possible that the effects observed in this study were due to the combined effects of nutrients present in bean sprouts, primarily coumestrol.

In mice, the oral ingestion of bean sprouts led to an approximately 10-fold increase in testosterone levels. In contrast, the results of the present study demonstrated a 1.5-fold increase in testosterone levels in human male participants. In previous experiments, mice typically consumed about 4 g of food daily and were provided with a diet containing 15% bean sprout powder. This level of intake effectively suppressed the development of colon cancer in *Apc^Min/+^* mice and Alzheimer’s disease pathology in 5XFAD mice [4,15]. A dose-dependent anticancer effect was observed in the *Apc^Min/+^* mouse model with bean sprout powder making up 5–15% of the total dietary intake. Based on the human equivalent dose (HDE) derived from mouse data, the estimated daily intake for humans would be 7.38 g of bean sprout powder, or approximately 25 capsules of 300 mg each [26]. In this study, however, a significant increase in testosterone level was observed with only 1/2.5 of this amount. These findings suggest that the testosterone-increasing effect may be dose-dependent, and higher levels of bean sprout ingestion may produce anticancer and anti-Alzheimer’s disease effects in humans similar to those observed in mice.

In summary, the results of the present study showed that the oral ingestion of bean sprouts can measurably increase serum testosterone levels in humans, suggesting that bean sprout consumption may be beneficial for managing LOH. Moreover, based on previous findings in animal models, a higher intake of bean sprouts may also show additional health benefits, such as anticancer and anti-Alzheimer’s effects, a reduction in visceral fat, and intestinal elongation.

## 4. Conclusions

Coumestrol-rich bean sprouts are effective as a dietary intervention for the treatment of LOH.

## Figures and Tables

**Figure 1 biology-14-00907-f001:**
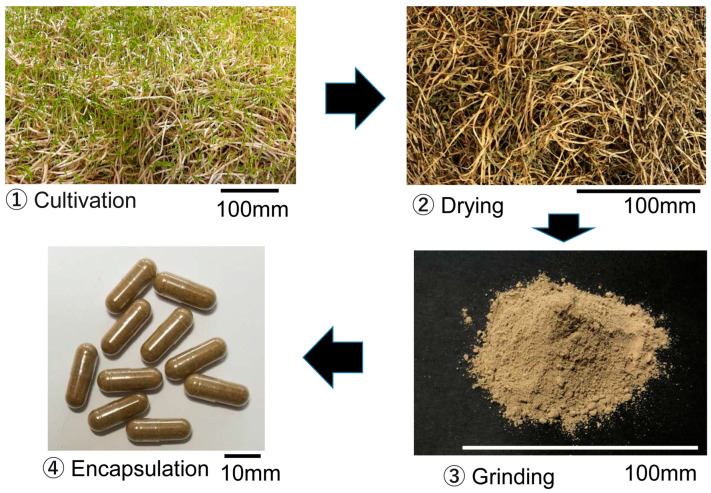
Sprout powder capsules. Mung bean sprouts were cultivated, dried, milled, packed into cellulose capsules, and provided to participants.

**Figure 2 biology-14-00907-f002:**
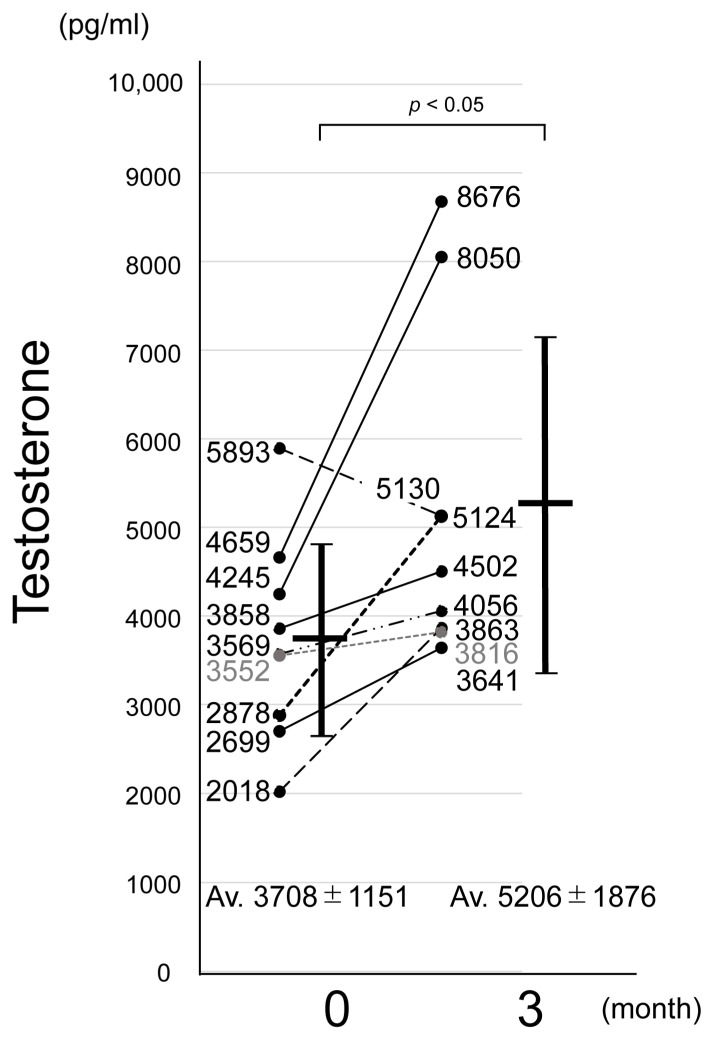
Time course of changes in total blood testosterone after ingestion of soybean sprouts. The vertical axis indicates total blood testosterone level (pg/mL), and the horizontal axis shows the period of bean sprout ingestion: baseline (pre-ingestion) and 3 months. Numerical values on the bars indicate mean ± standard deviation testosterone levels. Statistically significant differences were observed between periods (*p* < 0.05).

## Data Availability

The original contributions presented in this study are included in the article. Further inquiries can be directed to the corresponding author.

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
