# Peer review of "Oral Ingestion of Bean Sprouts Containing the HASPIN Inhibitor Coumestrol Increased Blood Testosterone Levels in Men"

_biology, 2025, doi:10.3390/biology14080907_

Round 1

Reviewer 1 Report

Comments and Suggestions for Authors

Introduction Section: Please focus this section on LOH, the mechanism of action of what triggers its occurrence in relation to the enzymes/substrates in testosterone biosynthesis. Please indicate what the objective of this study is.

Materials and methods section: The design in this study indicates collection data (blood testosterone) from the participants across time. Which means that treatment x time interaction is expected. There is nowhere interaction is reflected in the result section. The statistical tools used in analyzing collected data in this study are wrong. ANOVA with repeated measure should be used. Please seek the help of a biostatistician. Further, this study would have been cleaner if a separate control group (placebo group) was included. 

Results and discussion section: Because the statistical tools used for analyzing collected data are inappropriate, interpretation of analyzed is flawed and therefore renders the discussion of analyzed data weak. Further, please institute more mechanistic explanations in the discussion section.

Author Response

To Referee: 1

Thank you for finding our research worth and the valuable suggestions. We have added the expert member of statistical analysis to our study and revised our manuscript accordingly.

>Introduction Section: Please focus this section on LOH, the mechanism of action of what >triggers its occurrence in relation to the enzymes/substrates in testosterone biosynthesis. >Please indicate what the objective of this study is.

We agree with reviewers’ suggestion. We added sentences and reference #2  in Introduction Section

Line 46-62

LOH syndrome is a form of hypogonadism not attributed to chromosomal or genetic abnormalities, testicular injury, tumors, functional impairments, or central nervous system disorders. It primarily results from an age- or stress-induced decline in testosterone levels, with age-elated decrease in testosterone levels being the predominant cause. Treatment for LOH typically includes Chinese herbal medicine and testosterone replacement therapy (TRT), along with lifestyle modifications [2]. Testosterone is primarily produced by Leydig cells in the testes, with minor contributions from the adrenal cortex. The age-associated decline in testosterone levels is believed to result from reduced production and secretion by Leydig cells. In our previous work, we demonstrated that intraperitoneal administration of CHR-6494, an inhibitor of the serine-threonine kinase HASPIN, in ApcMin/+ mice—a model of familial colorectal cancer—inhibited both the development of colorectal cancer and testicular atrophy due to impaired spermatogenesis, while restoring blood testosterone levels [3]. Moreover, we found that oral administration of coumestrol, a natural HASPIN inhibitor, suppressed Alzheimer's disease onset in 5XFAD mice, a model of the disease, and elevated serum testosterone levels in both these mice and wild-type controls [4]. These findings suggest that HASPIN plays a role in testosterone production and secretion by Leydig cells.

>Materials and methods section: The design in this study indicates collection data (blood >testosterone) from the participants across time. Which means that treatment x time interaction >is expected. There is nowhere interaction is reflected in the result section. The statistical tools >used in analyzing collected data in this study are wrong. ANOVA with repeated measure >should be used. Please seek the help of a biostatistician. Further, this study would have been >cleaner if a separate control group (placebo group) was included. 

>Results and discussion section: Because the statistical tools used for analyzing collected >data are inappropriate, interpretation of analyzed is flawed and therefore renders the >discussion of analyzed data weak. Further, please institute more mechanistic explanations in >the discussion section.

We agree with reviewers’ suggestion. However, we have no placebo group in this experiment.

We plan to include a placebo in the next large-scale analysis as the reviewer’s suggestion.

We re-analysis of our data with the expert member of statistical analysis (line 5, Seitaro Kamiya)and we added and changed sentences as reviewers’ suggestion in Materials and methods section and Results and discussion section.

(Line 122-127)

2.3. Regimen

Participants took 10 capsules orally every day for 3 months at a time convenient to them, corresponding to a daily intake of 2.5–3.0 g of bean sprout powder. Samples of approximately 2 mL of venous blood were collected from each participant between 9:00 and 10:30 am at two time points: prior to the start of capsule ingestion, and at 3 months after starting ingestion. No capsules were taken on the day of blood collection.

(Line 137-141)

2.5. Statistical analysis.         

Data are expressed as mean ± standard deviation. The normality of paired differences was assessed using the Shapiro–Wilk test. Differences between baseline and 3-month measurements were analyzed using the Wilcoxon signed-rank test. P-values < 0.05 were considered statistically significant.

Reviewer 2 Report

Comments and Suggestions for Authors

Comments and Suggestions for Authors 

This pilot study addresses an important issue in men's health by exploring the potential of coumestrol-rich bean sprouts as a dietary intervention for late-onset hypogonadism (LOH). The authors report an increase in serum testosterone levels following daily intake of bean sprout capsules. With several revisions and additional clarifications, the overall quality and impact of the manuscript could be substantially improved. Below are my detailed comments for consideration. 

Introduction 

  • The first paragraph begins with LOH but abruptly shifts to HASPIN, cancer, and Alzheimers disease in the last sentence. To improve the logical flow, the authors should cite more literature directly related to LOH before introducing HASPIN. 
  • The main focus of this study is the increase in blood testosterone levels in men and its relevance to improving LOH. However, the Introduction disproportionately discusses cancer and Alzheimers disease. The authors should expand the LOH-related context and clarify how HASPIN inhibition mechanistically connects to testosterone regulation.  
  • The description of the test material is insufficient. Additional detail regarding the selection and characterization of coumestrol as the key marker compound is needed.  

Materials and Methods 

  • The sample size is extremely smallonly nine participants. This must be explicitly acknowledged as a major limitation in the Discussion, and the authors should stress the need for replication in a larger, randomized trial. 
  • The reported content of coumestrol (10–20 mg/100 g) in the manuscript is too broad. If analytical data for the actual materials used in the experiment are available, please provide the precise concentration of coumestrol. 
  • Participants took 10 capsules orally every day for three months. This is a substantial amount, making it impractical for general consumers. The authors should consider developing a liquid formulation for better consumer compliance. 
  • Only total testosterone was measured. Free testosterone or sex hormone-binding globulin (SHBG) levels would have provided a more accurate understanding of the physiological impact, particularly given the known variability in protein-bound testosterone fractions. If blood samples from the subjects are still available, I recommend conducting additional analyses. 

Results and Discussion 

  • The study does not account for potential confounding factors (e.g., physical activity, diet, sleep, alcohol intake) that may influence testosterone levels. A questionnaire or baseline control would have helped reduce these biases. If a questionnaire was used during the study, please include its contents in the manuscript for transparency and reproducibility. 
  • The study claims an increase in testosterone levels following oral intake of bean sprout capsules rich in coumestrol. However, no placebo or control group was included. This is a critical methodological flaw that undermines internal validity. Justifying this by referencing an unrelated food intervention study is scientifically inadequate. 
  • The authors suggest that coumestrol may act on Leydig cells via HASPIN inhibition, similar to synthetic HASPIN inhibitors. However, considering the relatively low coumestrol content (10–20 mg/100 g), this claim appears overstated. The possibility of synergistic effects with other phytochemicals should be discussed, and the need to identify the primary active compound(s) emphasized. Additionally, please cite other studies that have evaluated the pharmacological effects of coumestrol. This will help to discuss what dosage range of coumestrol might be considered effective in humans. 
  • The proposed mechanism involving HASPIN inhibition and Wnt/β-catenin signaling is speculative. No in vitro or in vivo mechanistic data are provided in this study. Citing previous mouse studies does not sufficiently support mechanistic claims in humans. Bridging studies or at least indirect biomarker assessments are required for such extrapolations. The dataset presented in this study is rather limited. Therefore, I strongly suggest incorporating more detailed references to previous in vivo studies to provide readers with a better scientific context. 
  • Health claims such as antiaging, anticancer, and anti-Alzheimers effects are not supported by the current data. Such statements are inappropriate in a pilot study involving nine subjects, with no disease-related biomarkers or endpoints measured. 
  • The extrapolation from mice to humans is oversimplified and based solely on food weight. The authors did not apply a body surface area correction, which is the standard method for dose translation across species. A recalculation based on this model should be included to justify the chosen human dose. 

Figures 

  • In Figure 2, the font sizes of the Y-axis and X-axis labels are inconsistent and should be unified for visual consistency. Even if statistical significance was not achieved, please report the p-values. It would also be helpful to add a comment discussing whether the results might have reached significance if the sample size had been larger. 

Author Response

To Referee: 2

Thank you for finding our research worth and the valuable suggestions. We have added the expert member of statistical analysis to our study and revised our manuscript accordingly.

We added and changed sentences as reviewers’ suggestion.

>Comments and Suggestions for Authors 

This pilot study addresses an important issue in men's health by exploring the potential of coumestrol-rich bean sprouts as a dietary intervention for late-onset hypogonadism (LOH). The authors report an increase in serum testosterone levels following daily intake of bean sprout capsules. With several revisions and additional clarifications, the overall quality and impact of the manuscript could be substantially improved. Below are my detailed comments for consideration. 

>Introduction 

  • The first paragraph begins with LOH but abruptly shifts to HASPIN, cancer, and Alzheimer’s disease in the last sentence. To improve the logical flow, the authors should cite more literature directly related to LOH before introducing HASPIN. 
  • The main focus of this study is the increase in blood testosterone levels in men and its relevance to improving LOH. However, the Introduction disproportionately discusses cancer and Alzheimer’s disease. The authors should expand the LOH-related context and clarify how HASPIN inhibition mechanistically connects to testosterone regulation.  
  • The description of the test material is insufficient. Additional detail regarding the selection and characterization of coumestrol as the key marker compound is needed.  

We added and changed sentences, and added reference #2 as reviewers’ suggestion.

Line 46-62

LOH syndrome is a form of hypogonadism not attributed to chromosomal or genetic abnormalities, testicular injury, tumors, functional impairments, or central nervous system disorders. It primarily results from an age- or stress-induced decline in testosterone levels, with age-elated decrease in testosterone levels being the predominant cause. Treatment for LOH typically includes Chinese herbal medicine and testosterone replacement therapy (TRT), along with lifestyle modifications [2]. Testosterone is primarily produced by Leydig cells in the testes, with minor contributions from the adrenal cortex. The age-associated decline in testosterone levels is believed to result from reduced production and secretion by Leydig cells. In our previous work, we demonstrated that intraperitoneal administration of CHR-6494, an inhibitor of the serine-threonine kinase HASPIN, in ApcMin/+ mice—a model of familial colorectal cancer—inhibited both the development of colorectal cancer and testicular atrophy due to impaired spermatogenesis, while restoring blood testosterone levels [3]. Moreover, we found that oral administration of coumestrol, a natural HASPIN inhibitor, suppressed Alzheimer's disease onset in 5XFAD mice, a model of the disease, and elevated serum testosterone levels in both these mice and wild-type controls [4]. These findings suggest that HASPIN plays a role in testosterone production and secretion by Leydig cells.

>Materials and Methods 

  • The sample size is extremely small—only nine participants. This must be explicitly acknowledged as a major limitation in the Discussion, and the authors should stress the need for replication in a larger, randomized trial. 
  •  

We added and changed sentences in Results and Discussion as reviewers’ suggestion.

Line 155-158

Although this small-scale study in elderly individuals demonstrated a significant increase in testosterone levels, larger studies involving a broader age range are needed to further clarify the testosterone-boosting effects of bean sprouts and to identify the target populations most likely to benefit.

  • The reported content of coumestrol (10–20 mg/100 g) in the manuscript is too broad. If analytical data for the actual materials used in the experiment are available, please provide the precise concentration of coumestrol. 

We rewrote a sentence to the best of our knowledge as reviewers’ suggestion.

Line 112

Bean sprouts (Vigna radiata) containing 14 mg of coumestrol per 100 g of fresh weight were cultivated as described previously [14].

  • Participants took 10 capsules orally every day for three months. This is a substantial amount, making it impractical for general consumers. The authors should consider developing a liquid formulation for better consumer compliance. 

Thank you for your suggestion, we are developing a compact pill.

  • Only total testosterone was measured. Free testosterone or sex hormone-binding globulin (SHBG) levels would have provided a more accurate understanding of the physiological impact, particularly given the known variability in protein-bound testosterone fractions. If blood samples from the subjects are still available, I recommend conducting additional analyses. 

Thank you for your suggestion, we plan to include a placebo in the next large-scale analysis with analysis of free testosterone or sex hormone-binding globulin (SHBG) levels as the reviewer’s suggestion.

We added sentences Line 163, 164,

Although testosterone subtypes were not assessed separately in this study,

Results and Discussion 

  • The study does not account for potential confounding factors (e.g., physical activity, diet, sleep, alcohol intake) that may influence testosterone levels. A questionnaire or baseline control would have helped reduce these biases. If a questionnaire was used during the study, please include its contents in the manuscript for transparency and reproducibility. 
  • The study claims an increase in testosterone levels following oral intake of bean sprout capsules rich in coumestrol. However, no placebo or control group was included. This is a critical methodological flaw that undermines internal validity. Justifying this by referencing an unrelated food intervention study is scientifically inadequate. 

We agree with reviewers’ suggestion. However, we have no placebo group in this experiment.

In this study, we asked subjects to eat bean sprouts without any special lifestyle changes. One out of nine subjects experienced a drop in testosterone levels after three months, which may have been influenced by factors such as physical activity, diet, sleep, and alcohol intake before the blood draw. We plan to include a placebo in the next large-scale analysis with analysis of free testosterone or sex hormone-binding globulin (SHBG) levels as the reviewer’s suggestion.

We agree with reviewers’ suggestion.  

We rewrote a sentence to the best of our knowledge as reviewers’ suggestion.

Line 155-158

Given that eight out of nine participants showed an increase in testosterone levels, it is unlikely that the observed effect was due to placebo alone. Participants were asked to consume the bean sprouts without making any specific lifestyle changes. One of the nine subjects experienced a decrease in testosterone levels after three months, which may have been influenced by factors such as physical activity, diet, sleep, or alcohol intake prior to blood sampling. Therefore, the observed increase in blood testosterone level was attributed to the oral ingestion of bean sprouts rather than any placebo effect. Although this small-scale study in elderly individuals demonstrated a significant increase in testosterone levels, larger studies involving a broader age range are needed to further clarify the testosterone-boosting effects of bean sprouts and to identify the target populations most likely to benefit.

  • The authors suggest that coumestrol may act on Leydig cells via HASPIN inhibition, similar to synthetic HASPIN inhibitors. However, considering the relatively low coumestrol content (10–20 mg/100 g), this claim appears overstated. The possibility of synergistic effects with other phytochemicals should be discussed, and the need to identify the primary active compound(s) emphasized. Additionally, please cite other studies that have evaluated the pharmacological effects of coumestrol. This will help to discuss what dosage range of coumestrol might be considered effective in humans. 

We agree with reviewers’ suggestion. We re-wrote our discussion and added reference # 25.

Line 177-1181

Soybeans and bean sprouts contain various polyphenols; however, unsprouted soybeans contain very little coumestrol, which is found mainly in bean sprouts [14]. Coumestrol is a polyphenol phytoestrogen that has been shown to bind to estrogen receptors [24]. Phytoestrogens, including coumestrol, can exert biological effects even at very low concentrations and are categorized as natural endocrine-disrupting chemicals [25].

Line 185-190

Furthermore, coumestrol may regulate testosterone production via multiple pathways. Although the phytoestrogenic effects of polyphenols from soy-based foods have been documented, no prior studies have demonstrated an increase in testosterone levels associated with their intake. Therefore, it is also possible that the effects observed in this study were due to the combined effects of nutrients present in bean sprouts, primarily coumestrol.

  • The proposed mechanism involving HASPIN inhibition and Wnt/β-catenin signaling is speculative. No in vitro or in vivo mechanistic data are provided in this study. Citing previous mouse studies does not sufficiently support mechanistic claims in humans. Bridging studies or at least indirect biomarker assessments are required for such extrapolations. The dataset presented in this study is rather limited. Therefore, I strongly suggest incorporating more detailed references to previous in vivo studies to provide readers with a better scientific context. 

We agree with reviewers’ suggestion. We deleted the proposed mechanism involving HASPIN inhibition and Wnt/β-catenin signaling.

  • Health claims such as “antiaging,” “anticancer,” and “anti-Alzheimer’s effects” are not supported by the current data. Such statements are inappropriate in a pilot study involving nine subjects, with no disease-related biomarkers or endpoints measured. 

We agree with reviewers’ suggestion. We deleted the endpoints of “antiaging,” “anticancer,” and “anti-Alzheimer’s effects” 

  • The extrapolation from mice to humans is oversimplified and based solely on food weight. The authors did not apply a body surface area correction, which is the standard method for dose translation across species. A recalculation based on this model should be included to justify the chosen human dose. 

 We agree with reviewers’ suggestion. We added sentences and reference #7 in discussion” 

Line 202-208

Alternatively, based on the human equivalent dose (HDE) derived from mouse data, the estimated daily intake for humans would be 7.38 g of bean sprout powder, or approximately 25 capsules of 300 mg each [27]. In this study, however, a significant increase in testosterone level was observed with only 1/20 to 1/2.5 of this amount. These findings suggest that the testosterone-increasing effect may be dose-dependent, and higher levels of bean sprout ingestion may produce anticancer and anti-Alzheimer’s disease effects in humans similar to those observed in mice.

Figures 

  • In Figure 2, the font sizes of the Y-axis and X-axis labels are inconsistent and should be unified for visual consistency. Even if statistical significance was not achieved, please report the p-values. It would also be helpful to add a comment discussing whether the results might have reached significance if the sample size had been larger. 

We added and changed sentences as reviewers’ suggestion.

Line 137-141

2.5. Statistical analysis.           

Data are expressed as mean ± standard deviation. The normality of paired differences was assessed using the Shapiro–Wilk test. Differences between baseline and 3-month measurements were analyzed using the Wilcoxon signed-rank test. P-values < 0.05 were considered statistically significant.

Line 219-220

Statistically significant differences were observed between periods (p < 0.05).

Reviewer 3 Report

Comments and Suggestions for Authors

The MS entitled “Oral Ingestion of Bean Sprouts…” investigated the effect of dietary bean sprouts powder at a dosage of 2.5–3.0 g daily for 3 months on serum testosterone level in 9 male volunteers. In my opinion, the most problem is that the presented content is not enough for the publication in Biology.

General comment

This study designed at a simple way with a single result of Testosterone by ELISA.

The figures in the result part are repeated, the figure presentation is not in the formal way, and the statics of the results is not clear.

The introduction is not logically writing.

The discuss should include the possible reason for unchanged participant, as authors showed in L121-122 that the serum… eight of nine participants compared to baseline.

Specific comment

L43-44   sentences are not logically, here need to point out the potential causes related to late-onset hypogonadism (LOH) to bring out HASPIN, then introduce the potential role or the research progress of HASPIN; not directly cancer and aging-related diseases.

L107 ELISA kit parameter should be included such as sensitivity and coefficient of variation.

L115 In this study, there are only two groups (time points or stages), so it is not correct by using Student’s t test and one-way analysis of variance.

Author Response

To Referee: 3

Thank you for the valuable suggestions.We have added the expert member of statistical analysis to our study and revised our manuscript accordingly. We added and changed sentences as reviewers’ suggestion.

>General comment

>This study designed at a simple way with a single result of Testosterone by ELISA.

>The figures in the result part are repeated, the figure presentation is not in the formal way, and

>the statics of the results is not clear.

Thank you for the valuable suggestions.

I am sorry. There were overlapping figures, so I removed one.

We agree with reviewers’ comment. we made the vertical axis format of the figure the same as the others to make it easier to read.

We re-analysis of our data with the expert member of statistical analysis and we added and changed sentences as reviewers’ suggestion.

>The introduction is not logically writing.

We agree with reviewers’ comment. We added and changed sentences as reviewers’ suggestion

Line 46-62

LOH syndrome is a form of hypogonadism not attributed to chromosomal or genetic abnormalities, testicular injury, tumors, functional impairments, or central nervous system disorders. It primarily results from an age- or stress-induced decline in testosterone levels, with age-elated decrease in testosterone levels being the predominant cause. Treatment for LOH typically includes Chinese herbal medicine and testosterone replacement therapy (TRT), along with lifestyle modifications [2]. Testosterone is primarily produced by Leydig cells in the testes, with minor contributions from the adrenal cortex. The age-associated decline in testosterone levels is believed to result from reduced production and secretion by Leydig cells. In our previous work, we demonstrated that intraperitoneal administration of CHR-6494, an inhibitor of the serine-threonine kinase HASPIN, in ApcMin/+ mice—a model of familial colorectal cancer—inhibited both the development of colorectal cancer and testicular atrophy due to impaired spermatogenesis, while restoring blood testosterone levels [3]. Moreover, we found that oral administration of coumestrol, a natural HASPIN inhibitor, suppressed Alzheimer's disease onset in 5XFAD mice, a model of the disease, and elevated serum testosterone levels in both these mice and wild-type controls [4]. These findings suggest that HASPIN plays a role in testosterone production and secretion by Leydig cells.

>The discuss should include the possible reason for unchanged participant, as authors showed >in L121-122 that the serum… eight of nine participants compared to baseline.

 We agree with reviewers’ comment. We added and changed sentences as reviewers’ suggestion.

Line 148-158

Given that eight out of nine participants showed an increase in testosterone levels, it is unlikely that the observed effect was due to placebo alone. Participants were asked to consume the bean sprouts without making any specific lifestyle changes. One of the nine subjects experienced a decrease in testosterone levels after three months, which may have been influenced by factors such as physical activity, diet, sleep, or alcohol intake prior to blood sampling. Therefore, the observed increase in blood testosterone level was attributed to the oral ingestion of bean sprouts rather than any placebo effect. Although this small-scale study in elderly individuals demonstrated a significant increase in testosterone levels, larger studies involving a broader age range are needed to further clarify the testosterone-boosting effects of bean sprouts and to identify the target populations most likely to benefit.

>Specific comment

>L43-44   sentences are not logically, here need to point out the potential causes related to late-onset hypogonadism (LOH) to bring out HASPIN, then introduce the potential role or the research progress of HASPIN; not directly cancer and aging-related diseases.

 We agree with reviewers’ comment. We removed the sentences as reviewers’ suggestion

>L107 ELISA kit parameter should be included such as sensitivity and coefficient of variation.

Line 133-134

We added sentences in materials and methods as reviewers’ suggestion.

The detection sensitivity of the assay was 2.6 pg/mL, with an average coefficient of variation of 3.3% across sample concentrations.

>L115 In this study, there are only two groups (time points or stages), so it is not correct by >using Student’s test and one-way analysis of variance.

We have added the expert member of statistical analysis to our study and revised our manuscript accordingly. We added and changed sentences as reviewers’ suggestion.

We re-analysis of our data with the expert member of statistical analysis (line 5, Seitaro Kamiya)and we added and changed sentences as reviewers’ suggestion in Materials and methods section and Results and discussion section.

Line 137-141

2.5. Statistical analysis.         

Data are expressed as mean ± standard deviation. The normality of paired differences was assessed using the Shapiro–Wilk test. Differences between baseline and 3-month measurements were analyzed using the Wilcoxon signed-rank test. P-values < 0.05 were considered statistically significant.

Round 2

Reviewer 2 Report

Comments and Suggestions for Authors

The revised manuscript shows significant improvement compared to the previous version. If the following points are further addressed, it will provide readers with a clearer understanding.

Results and Discussion:

 The conversion based on HED in Lines 203–209 has been adequately revised. However, the estimation related to the intake of 200 capsules in Lines 194–203 remains inappropriate, as it is based solely on a simple body weight comparison between humans and mice. A proper correction is necessary.

Figure:

 The font sizes of the X- and Y-axis labels are still inconsistent. For improved readability, they must be unified. In addition, although the p-values have been included in the figure legend, they are not shown in the figure itself. It is recommended to add the p-values directly to the figure as well.

Author Response

To Reviewer 2

I am truly grateful to take your time for us and for the valuable suggestions.

The revised manuscript shows significant improvement compared to the previous version. If the following points are further addressed, it will provide readers with a clearer understanding.

Results and Discussion:

 The conversion based on HED in Lines 203–209 has been adequately revised. However, the estimation related to the intake of 200 capsules in Lines 194–203 remains inappropriate, as it is based solely on a simple body weight comparison between humans and mice. A proper correction is necessary.

We agree with reviewers’ comment. We removed the sentence of line 198-202.

In mice, oral ingestion of bean sprouts led to an approximately 10-fold increase in testosterone levels. In contrast, the results of the present study demonstrated a 1.5-fold increase in testosterone levels in human male participants. In previous experiments, mice typically consumed about 4 g of food daily and were provided a diet containing 15% bean sprout powder. This level of intake effectively suppressed the development of colon cancer in ApcMin/+ mice and Alzheimer’s disease pathology in 5XFAD mice [4, 15]. A dose-dependent anticancer effect was observed in the ApcMin/+ mouse model with bean sprout powder making up 5%–15% of the total dietary intake (data not shown). Humans consume approximately 400 g of carbohydrates daily, corresponding to roughly 100 times the intake of mice [26]. To match the relative intake used in the mouse studies, humans would need to consume about 60 g of bean sprout powder daily, equivalent to approximately 200 capsules each containing 300 mg per day. Alternatively, Based on the human equivalent dose (HDE) derived from mouse data, the estimated daily intake for humans would be 7.38 g of bean sprout powder, or approximately 25 capsules of 300 mg each [27]. In this study, however, a significant increase in testosterone level was observed with only 1/20 to 1/2.5 of this amount. These findings suggest that the testosterone-increasing effect may be dose-dependent, and higher levels of bean sprout ingestion may produce anticancer and anti-Alzheimer’s disease effects in humans similar to those observed in mice.

Figure:

 The font sizes of the X- and Y-axis labels are still inconsistent. For improved readability, they must be unified. In addition, although the p-values have been included in the figure legend, they are not shown in the figure itself. It is recommended to add the p-values directly to the figure as well.

We agree with reviewers’ comment. We corrected the figure 2.

The font size will be left to the publisher, so We will also attach the PowerPoint file to the publisher.

Reviewer 3 Report

Comments and Suggestions for Authors

Authors have revised the brief report appropriately.

Author Response

To Reviewer 3

Thank you for taking your time for us and for the valuable suggestions.

I am truly grateful to you.